# Emerging Extracellular Molecular Targets for Innovative Pharmacological Approaches to Resistant Mtb Infection

**DOI:** 10.3390/biom13060999

**Published:** 2023-06-16

**Authors:** Alice Italia, Mohammed Monsoor Shaik, Francesco Peri

**Affiliations:** Department of Biotechnology and Biosciences, University of Milano-Bicocca, 20126 Milano, Italy; a.italia@campus.unimib.it (A.I.); md.monsoor@gmail.com (M.M.S.)

**Keywords:** infectious diseases, tuberculosis, drug development, virulence factors, immune system

## Abstract

Emerging pharmacological strategies that target major virulence factors of antibiotic-resistant Mycobacterium tuberculosis (Mtb) are presented and discussed. This review is divided into three parts corresponding to structures and functions important for Mtb pathogenicity: the cell wall, the lipoarabinomannan, and the secretory proteins. Within the cell wall, we further focus on three biopolymeric sub-components: mycolic acids, arabinogalactan, and peptidoglycan. We present a comprehensive overview of drugs and drug candidates that target cell walls, envelopes, and secretory systems. An understanding at a molecular level of Mtb pathogenesis is provided, and potential future directions in therapeutic strategies are suggested to access new drugs to combat the growing global threat of antibiotic-resistant Mtb infection.

## 1. Introduction

Mycobacterium tuberculosis (Mtb) is the agent that causes tuberculosis (TB), a disease that mostly affects the lungs [1,2]. Mtb is a persistent public health threat, with approximately 28.000 individuals developing the disease and 4100 deaths per day (WHO 2022) [3]. It employs diverse virulence strategies, such as secreting proteins to interfere with host signaling and manipulating host immune response [2,4,5]. Mtb adapts by accumulating polymorphisms throughout the treatment, leading to drug resistance, including multidrug resistance (MDR-TB) and extensive drug resistance (XDR-TB) [6,7,8]. Given the pathogen’s complexity, a better understanding at a molecular level of the host-pathogen interactions and the mechanism of escape could provide new targets for the development of drugs and vaccines to prevent Mtb diffusion and overcome drug resistance. The different virulence factors of Mtb and current advances in their targeting are reviewed with a focus on therapeutic approaches targeting the cell wall components (mycolic acids, arabinogalactan, peptidoglycan), lipoarabinomannan, and the secretory systems [2,9,10]. 

The aim of this review is to report emerging pharmacological strategies for the development of drugs against Mtb infection deriving from the discovery of new targets. However, because of the wideness of the topic, we limited our analysis to drugs/molecules targeting the cell wall polymeric components and the secretory proteins. 

## 2. Targeting Cell Wall Biopolymers

The cell wall of Mtb is a complex tripartite structure consisting of mycolic acids, arabinogalactan and peptidoglycan, along with several heteropolymers [8]. These biopolymers contribute to the virulence of Mtb, playing a role in the resistance to host immune defenses and antimicrobial agents. The intricate organization of the cell envelope gives an intrinsically low permeability [8,11]. 

### 2.1. Mycolic Acids Biosynthesis

Mycolic acids (MA) are distinctive of the mycomembrane that is the major component of both the outer leaflet and inner leaflet (Figure 1). In the outer leaflet, MA can be free or attached to trehalose sugar to make trehalose mono-mycolate (TMM) or trehalose dimycolate (TDM). In the inner leaflet, MA are covalently attached to the underlying arabinogalactan of the arabinogalactan–peptidoglycan matrix [12]. MA are long-chain fatty acids (70–90 carbons) that make it impermeable and protect the bacterial cell from external stressors and phagocytic engulfment by host cells. Their synthesis is a complex process involving several enzymes, including Fatty Acid Synthase I and II (FAS I and FAS II) systems; the former is conserved in mammalian cells, whereas the latter is unique to mycobacteria. The conformation of the MA plays an important role in the resistance and strength of the Mtb cell wall [12,13]. Recent studies on the conformation of the MA layer identified that the so-called U conformation, with a single bend, is the most stable, while the W conformation, with two bends, is more labile [14]. It is suggested that the majority of MA are in U single-bend conformation, resulting in substantially thicker and denser membranes.

The synthesis of MA is crucial for the structural integrity of the Mtb cell wall, and its inhibition has been found to impede Mtb survival within macrophages. This highlights the significant role of MA in the pathogenicity of this bacterium. Many studies have been conducted to identify potential inhibitors that target the biosynthesis of MA. Among several enzymes acting in the biosynthesis of MA, three of them are considered effective targets of TB chemotherapy: the enoyl-acyl protein reductase (InhA), the beta-ketoacyl-ACP synthase A (KasA), and the mycobacterial membrane protein Large 3 (MmpL3).

InhA is a reductase that reduces NADH 2-trans-enoyl-ACP during the FASII elongation step, thereby enabling the stepwise elongation of fatty acid chains. InhA inhibition by isoniazid (compound n.2, Table 1) reduces Mtb survival. Emerging resistant strains to isoniazid treatment have shown mutations in InhA or in KatG, an Mtb enzyme that is involved in the formation of an active isoniazid metabolite [15]. The active isoniazid metabolite links covalently to NADH, thus inhibiting the NADH-dependent InhA activation. This leads to an accumulation of long-chain fatty acids inside the Mtb cytosol, inducing cytotoxicity [16]. 

Ethionamide, delamanid, and prothionamide (compounds n.9–11, Table 1) are well-known second-line anti-TB prodrugs. They are structurally analogous to the isoniazid and could be considered prodrugs that are activated by ethA-encoded mono-oxygenase to form adducts with NAD that subsequently block the action of fatty acid synthase [16,21,24]. It was shown that most InhA inhibitors bind to its catalytic site, also interacting with NAD cofactor and with Tyr158. Interestingly, the benzimidazole and benzimidazole-based InhA inhibitors bind the enzyme in an extended hydrophobic pocket [16]. 

During the FASII elongation, KasA covers a crucial role, and its inhibition leads to the inhibition of growth and bacterial cell lysis. Abrahams et al. (Glaxo Smithkline) identified GSK3011724A (compound n.7, Table 1) as a powerful inhibitor of KasA. They showed the specificity of this new compound and discovered that it binds within the large acyl channel of KasA, thus blocking the enzyme in the open conformation and preventing its activity [25]. It has also been recently reported that GSK3011724A strongly augmented the isoniazid effect, thus resulting in a synergistic antibiotic effect [18]. Ramesh et al., through in vitro and in silico analyses, additionally identified indole chalcone derivatives as KasA inhibitors, in particular, the compound (*E*)-1-(furan-3-yl)-3-(1*H*-indol-3-yl)prop-2-en-1-one (compound n.8, Table 1) showed strong binding to KasA and potent anti-tubercular activity without showing toxicity to human and murine cells [19].

MmpL3 is involved in the transport and production of TMM, which is required for the binding of mycolates to the arabinogalactan layer during cell wall synthesis. The impairment of this transport induces an accumulation of trehalose in the cytoplasm and a weakening of the cell membrane’s permeability. In the last decade, different small molecules were proposed as potential inhibitors of MmpL3. AU1235 (urea derivative), BM212 (pyrrole derivative), and SQ109 (compounds n.4-6, Table 1) are currently under clinical trials and have been shown to inhibit the MmpL3 in preclinical studies [23].

In computational studies, Cytosporone E and its analogs (compound n.12, Table 1) have been identified to inhibit InhA, KasA, and MmpI3. Cytosporone E derivatives have been demonstrated to interact with some residues in the InhA binding site similarly to InhA inhibitors such as triclosan, mandelic acid derivatives, and carfilzomib. Cytosporone E analogs interact through hydrogen bonds with Thr313 and Thr315 of KasA and through an additional hydrogen bond with His311 located in the catalytic triad of KasA. These multiple interactions lead to the highest target affinity among the other inhibitors tested. The ligands which present a high docking score for binding InhA also have significant binding affinity for MmpL3 and interact with residues Asp256, Ile297, Phe260, and Leu642 in the binding pocket of MmpL3 [23]. 

The quinoline-isoniazid conjugates (compounds n.3, Table 1) were reported to be potential anti-TB compounds. These compounds are synthesized by the condensation of triazolylquinolines with isoniazid by heating. Quinoline-isoniazid hybrids inhibit InhA action at minimal concentrations in the 0.25–0.50 μg/mL range and are bactericidal in vitro. These compounds are also well tolerated by mammalian cells, suggesting that they are promising drug hits [15].

Fad32, an enzyme involved in mycolic acid biosynthesis, has been shown to be overexpressed during Mtb’s active growth phase and underexpressed during the latent phase. Thus, inhibiting Fad32 might be a potential strategy for targeting Mtb during the growth phase. Modak et al. studied the role of diaryl coumarins (compound n.1, Table 1) in the inhibition of Fad32 and observed a decreased Mtb growth rate during the active growth state [17]. 

### 2.2. Arabinogalactan (AG) Biosynthesis

The arabinogalactan (AG) layer is the middle layer of the cell wall core, and it is attached to the *N*-acetylmuramic acid of the peptidoglycan through the rhamnose–GlcNAc disaccharide linker (Figure 1). The structure’s complexity is then increased by the presence of galactose residues in the galactan linked to the branching arabinose chains [12]. The galactosamine (GalN) unit in the arabino domain plays a role in the physio-pathology of Mtb. Studies on its deletion have been correlated with a decrease in Mtb virulence. GalN inhibits the innate immune response of the host by inhibiting the maturation and activation of human peripheral blood monocyte-derived dendritic cells (hPMC-DCs). It also plays a role in DC-SIGN affinity, stimulates IL-10 production, and inhibits the NF-kB TLR2-dependent pathway [26]. Despite GalN's role in Mtb virulence, at the moment, there are no compounds available that target this sugar. This, in our opinion, is a potentially interesting target for future drug discovery.

The structural stability of the cell wall is provided by several different variants (chemotypes) of AG, which differ in size, branching, and substitution patterns. The synthesis of AG is mediated by a set of arabinosyltransferases, which are essential for bacterial survival [12]. EmbA, EmbB, and EmbC are essential arabinosyltrasferases for arabinogalactan synthesis and are the primary target of ethambutol, a first-line antibiotic that inhibits AG biosynthesis [16]. Zhang et al. elucidated the structure and role of EMbB- AcpM along with ethambutol via cryo-EM [27,28,29,30,31]. Another emerging and interesting drug target is the DprE1 enzyme, which is involved in arabinogalactan biosynthesis. There are several molecules that inhibit its activity, and the most promising are benzothiazinones BTZ043 and PBTZ169 (compounds n.13-14, Table 2), both of which are currently in or have completed phase IIa clinical trials [16]. 

It is worth noting that these molecules have synergistic action with other anti-TB drugs, such as bedaquiline (Sirturo), which have been shown to increase the permeability of the cell wall. Additionally, using a genome-scale CRISPRi screen, Poulton et al. determined the sensitivity and resistance to PBTZ169 as well as dozens of hit genes that might be targeted to increase the BTZs activity [33].

Caprazene (CPZ) (compound n.15, Table 2) has been shown to inhibit the arabinogalactan biosynthesis enzyme translocase I (MurX), while its derivative CPZEN 45 (n.16, Table 2) inhibits the phospho-*N*-acetyl glucosaminyl transferase (WecA) [32]. WecA is essential in the transformation of UDP-GlcNAc to decaprenyl-P-P-GlcNAc, which is a component of mycolylarabinogalactan in Mtb. Thus, inhibition of WecA blocks cell wall biosynthesis, leading to the death of the pathogen [34]. CPZEN-45 has been shown to be more active than the natural precursor CPZ as well as more cytotoxic to the host cells. Thus, work is in progress to find a modification of CPZEN-45 that is less toxic while retaining the high activity/selectivity [32]. 

### 2.3. Peptidoglycan Layer

The peptidoglycan layer, which consists of covalently linked sugars and amino acids, provides shape and rigidity to Mtb (Figure 1). Structurally, it consists of linear glycan chains containing alternated units of N-acetylglucosamine (GlcNAc) and N-acetylmuramic acid (MurNAc), cross-linked by a peptidic bridge. It is unique to bacterial cells, and the enzymes that catalyze its biosynthesis are essential and therefore offer an attractive target for new antibiotics against TB [35]. It has been reported that peptidoglycan cell wall components can interfere with the host’s immune response and inhibit macrophage responses to IFN- γ at the transcription level [36,37,38,39]. Peptidoglycan synthesis is a complex biosynthesis composed of several steps involving different enzymes at both the cytoplasmic and membrane levels. Cytoplasmatic enzymes such as Glm, Mur, Alanine Racemase (Alr) and D-Ala-D-Ala-ligase (Ddl) are better characterized at the functional level than at the structural level. These enzymes are essential and important pharmacological targets [35]. 

The Alr and the Ddl are required for the building block formation of the peptidoglycan. Thus, the inhibition of these two actors results in the impairment of the biosynthesis of peptidoglycans with anti-TB activity. D-cycloserine (DCS) (n.17, Table 3), an antibiotic included in the second-line anti-TB drug medication, has structural analogs to D-Ala [40,41,42,43]. It has been proven to inhibit Ddl and Alr via D-Ala binding sites in the same way that D-Ala does [42,44]. Several studies were conducted to identify novel inhibitors of Mur enzymes. Enyan et al. recently identified in an in vitro study seven compounds with greater than 50% inhibition of MurB and MurE enzymes [45]. Additionally, the MurB inhibitors Sulfadoxine and Pyrimethamine (n.18-19, Table 3) were discovered in a repurposing screening, while Lifitegrast and Sildenafil (n.21-20, Table 3), which have shown the most reliable interactions with the MurE, might also be further explored for the repurposing [46].

Through different computational analyses and molecular dynamics simulations, four compounds were identified as potential MurD inhibitors. These four compounds (ZINC11881196, ZINC12247644, ZINC14995379, and PubChem6185; n.22-25, Table 3) showed the ability to form stable, energetically favored complexes with MurD, making them potential inhibitors [48]. Even though the inhibitors targeting the above-mentioned enzyme still did not reach the clinical phase of development, we are, however, convinced that these computational studies provide a foundation for the synthesis and optimization of novel anti-TB drugs.

## 3. Lipoarabinomannan (LAM) Biosynthesis

LAM is one of the major mycobacterial cell envelope-associated glycolipids and is a potent immunomodulator. It has been shown to inhibit the host immune response, promote bacterial survival, and contribute to the establishment of chronic infections. It is involved in the control of apoptosis-induced macrophages, hampering the production of IL-12 by DCs. LAM also acts in the dephosphorylation of tyrosine kinase and inactivates the kinase activity and, eventually, the function of T-cells and phagocytic cells. Additionally, it was shown to alter the signaling activities via its self-insertion in the membrane and its association with Toll-like Receptors (TLRs) and block calcium signaling to enhance Mtb survival [49]. 

Derivatives of LAM are ManLAM and AraLAM. ManLam is derived from the addition of a mannosyl cap at the terminal portion of LAM and is only found in slow-growing mycobacteria. AraLam is a polysaccharide that lacks the mannosyl cap (Figure 2) [35].

Innate immunity is essential to the host response against the Mtb invasion. LAM, ManLAm, and LM can be recognized by TLR2/TLR1 of the macrophages and DCs. These interactions induce cytokines production and granuloma formation. Dectin-2, a C-type lectin receptor, was likewise found to detect ManLAm and elicit a pro-inflammatory response by releasing IFN-γ and IL-2 [50]. 

LAM activates TLR-2 signaling, which inhibits MHC class II antigen presentation and allows infected antigen-presenting cells to function. This is associated with T cell evasion by T cells, and its presence in high concentrations during the chronic phase may shield static granulomas from immune system sterilization [49]. LAM and ManLAM are also involved in the escape through their interaction with DC-SIGN and the mannose receptor. The interactions with these receptors inhibit the release of pro-inflammatory cytokines such as IL-12, TNF-α, and IL-6 while boosting the release of anti-inflammatory IL-10 [51,52,53]. Importantly, ManLAM inhibits phagosome maturation once the bacteria are engulfed and allows the insertion of bacteria into macrophage membranes [54]. The binding of LAM to lactosylceramide (LacCer)-enriched lipid domains will lead to phagocytosis of the neutrophils. Nakayama et al. investigated the specificity of binding of several anti-LAM IgMs, such as TMDU3 and La066, which are directed against the mannan core of LAM and strongly bind to Mtb [55]. 

Furthermore, treatment with these antibodies has been shown to reduce the Mtb intracellular presence in human neutrophils. This indicates that the blockage of LacCer-enriched lipid micro-domains and LAM interaction can be a possible therapeutic and/or diagnostic target for Mtb infection [55]. Moreover, it was recently demonstrated that LAM could induce lipid droplet (LD) formation and activation of PPARγ involving host surface receptors, for instance, TRLs, CD36, CD14, and CD11b/Cd18. The inhibition of these LD formations can affect bacterial replication and cytokine production [56]. Thus, LAM and its derivatives have multiple roles as virulent factors and escape pathway activators as antigen factors that trigger the host immune system, resulting in a promising drug target [26,57].

Among the first-line treatments for TB, ethambutol (n.26, Table 4) acts by inhibiting the LAM synthesis during the static phase of the infection, affecting its immunomodulatory function, and modulating the structure of the Mtb cell wall [17]. LAM is also useful as a marker to test the presence of Mtb infection in urine using different types of antibodies against LAM; for example, U-LAM-based diagnostics in urine is a sensitive, non-invasive and rapid diagnostic test using an anti-LAM monoclonal antibody [58,59].

## 4. Targeting the Secretory Systems

Mtb evolved many secretory systems that are tightly regulated to deliver virulence factors and other effector proteins to the host cells. These secreted factors, in turn, modulate the host's immune responses through a plethora of strategies. Secretory proteins get localized in distinct cellular compartments and are exported by well-regulated secretion systems, which are important for their virulence and pathogenesis [60]. The distinctive features of the Mtb cell envelope necessitate the need for specific protein export requirements. Mycobacteria have four main protein export pathways, Tat, SecA, SecA2, and Type VII. Among these, there are two conserved systems that exist in all types of bacteria (Tat and SecA) and two specialized systems that exist in mycobacteria, corynebacteria, and a subset of low-GC Gram-positive bacteria (the SecA2 and type VII secretion pathways) (Figure 3) [61].

### 4.1. Sec Export Pathway

The SecA2 pathway is one of the specialized pathways that is involved in the Mtb pathogenicity and virulence. The SecA2-dependent protein export system is required for phagosome maturation arrest and, consequently, the growth of Mtb in macrophages. Among the virulence factors exported by SecA2, SapM phosphatase activity is required to prevent phagosome maturation in macrophages, and both SapM and PknG were found to contribute to Mtb replication in macrophages and inhibit the delivery to autophasolysosomes [62]. 

In Mtb, the SecA2 pathway is involved in activating the host Rig-I/MAVS signaling pathway and inducing an IFN-β response, suggesting a species-specificity of its action [63]. Additionally, it was recently shown that the SecA2 pathway activates the ATK kinase checkpoint protein, resulting in the formation of double-strand breaks (DBSs) in host DNA [64]. Several compounds have been identified as potential inhibitors of Sec pathways, including analogs of the dye rose Bengal, thiouracil analogs and thiazolidine derivatives [65,66,67]. 

Cui et al. investigated 23 rose Bengal derivatives (n.27, Table 5) as SecA inhibitors. Rose Bengal is a hydroxyxanthene dye, and several analogs have ATPase inhibitory actions and have been shown to have antibacterial properties via the SecA pathway [65]. Jin et al. analyzed the effects of the thiouracil analog, SCA-15 (n.28, Table 5), and have shown the inhibition of the ion-channel and ATPase functions of both SecA1 and SecA2. Efflux pumps play a crucial role in multi-drug resistance (MDR), often leading to suboptimal drug uptake. Interestingly, these compounds could also be interesting in overcoming the effect of efflux pumps that are responsible for MDR [66]. 

Different thiazolidine derivatives (n.29, Table 5) were tested for anti-mycobacterial activity. Khare et al. discovered Q30, M9, M26, U8, and R26 molecules as potent SecA2 inhibitors among 30 identified compounds [67]. These compounds have shown promising in vitro antimycobacterial activity and may open up a new path for the development of novel tuberculosis therapies. However, further research is necessary to optimize these lead compounds for clinical usage as well as to assess their efficacy in vivo. Overall, targeting Sec pathways may be a promising strategy for the development of new antimicrobial therapies for tuberculosis.

### 4.2. ESX Export Systems

ESX is another specialized transport system that is largely involved in the export of virulence factors across the cell envelope and is highly connected with the pathogenic potential of Mtb. It facilitates the secretion of the EsxA: EsxB virulence factors family or DNA [71,72]. It usually secretes substrates as a dimer, in which only one of the partners possesses an N-terminal helix-turn-helix (HTH) motif followed by the sequence motif YxxxD/E [72]. 

In Mtb, five encoded paralogous ESX systems (ESX-1 to ESX-5) were established, each with its own set of characteristics and functions. The ESX-1 is well-characterized for its role in the export of prototypic ESX proteins, including early secreted antigenic target 6 kDa (ESAT-6) and culture filtrate protein 10 kDa (CFP-10). These proteins are essential for Mtb pathogenicity and its ability to evade the host’s immune response [73,74,75]. 

ESX-1 also promotes Mtb escape from vacuoles into the host cell cytosol, thus promoting cell-to-cell spread. It does so in conjunction with the virulence factor ESAT-6, which directly contributes to the formation of pores in Mycobacterium-containing vacuole (MCV) membranes [74]. ESX-1 substrates have been linked to the inhibition of phagosome maturation and cytokine signaling by infected macrophages. ESX-1 interacts with TLR2 and inhibits the TLR signaling cascade facilitating the spread of bacteria via pore formation in mycobacterial phagosomes [75]. ESX-1 pathway is also involved in the early stages of infection, including the activation of the cytosolic signaling responses, targeting type I IFNs induction, immune response regulation, cell growth control, and apoptosis modulation [75]. ESX-1 is also involved in long-term Mtb infections by modulating the protein exports according to the host immune response [60,73,75]. Drever et al. 2021, identified a novel vulnerability in the ESX-1 system using inhibitors of protein synthesis and protein degradation (chloramphenicol, kanamycin and bortezomib) (n. 31-32-33, Table 5). These drugs specifically blocked the ESX-1 secretion and could be useful in designing new avenues of anti-TB drug targets [68].

ESX-3 is required for bacterial growth and is essential for the uptake of iron and zinc through mycobactin-mediated transport. It is responsible for the secretion of several proteins, including PE5-PPE4, which has a significant role in host-dependent siderophore-mediated iron-acquisition activity. This suggests that ESX-3 has a role in host defense mechanisms that limit iron availability [72,76]. In addition, it has recently been shown that inhibiting Salicylate synthase (MbtI) can impede iron absorption in Mtb and reduce its survival rate. This inhibition could be achieved with the administration of 5-(3-cyano-5-(trifluoromethyl) phenyl)furan-2-carboxylic acid (n.33, Table 5), paving the way for novel anti-tuberculosis therapies [69,77].

ESX-5 is exclusively found in slow-growing bacteria and is required for nutrient uptake and membrane permeability. According to evolutionary studies, the existence of ESX-5 is a relatively recent evolutionary trait. ESX-5 plays a significant role in host immunomodulation and immune evasion by secreting a wide array of substrates. These include the PE and PPE protein families, ESAT-6-like proteins, and the recently evolved PE_PGRS65 [75,78]. Recently novel 1,2,4-oxadiazole-containing compounds (n.34, Table 5) were identified to be able to inhibit ESX-5 secretion [70]. 

In conclusion, the ESX secretion systems are crucial for Mtb pathogenesis since they participate in virulence, metal homeostasis, and host-mycobacteria interaction. Understanding the molecular and cellular mechanisms of ESX systems and their secreted proteins is essential for developing new therapies to combat tuberculosis.

## 5. Conclusions

Mtb is a facultative, intracellular bacterium that persists in the host for an extended period of time and is the causative agent of tuberculosis diseases. Mtb has developed a vast variety of virulence strategies to avoid detection and coevolve with the host immune system. The virulence factors comprise metabolic enzymes, cell wall-associated proteins, and cell wall components. These virulence factors have evolved the ability to modify a variety of functions, including phagosome maturation, antigen presentation, and cytokine signaling. Considering the complexity and high impact of this pathogen on global health and the increment of MDR-TB and XDR-TB strains, here we have highlighted the importance of key virulence factors.

Novel therapeutics targeting the structural virulence factors of the Mtb belonging to the cell wall, the envelope and the secretory export systems have been presented. The different polymeric components of the cell wall play an important role in Mtb’s virulence and are attractive targets for developing new antibiotics against TB. We also presented the various secretory systems that evolved from Mtb with a focus on specialized pathways (ESX and SecA2), which are involved in evading macrophage cleanup and modulating the host immune response, providing a survival advantage. The review emphasizes the requirement for a better understanding of mycobacterial secretion systems and their involvement in structural and functional processes to design novel drugs against this persistent and lethal pathogen.

Understanding the mechanism of these pathogenic factors could lead to the identification and characterization of potent inhibitors to overcome Mtb’s drug resistance. This problem continues to be a major hurdle to controlling and combating this lethal illness and its spread. As a result, the discovery of novel drugs that target virulence factors might be a potential way to deliver effective therapies for TB while also mitigating the public health threat.

Future research should focus on a deeper understanding of molecular aspects of the complex virulence factors employed by Mtb, along with their intricate secretion systems. This knowledge will pave the way for the development of novel therapeutic strategies targeting these systems, leading to more effective therapies against Mtb. Additionally, investigating alternative levels of Mtb’s biology, such as the regulation of virulence factor expression or the interference with host-pathogen interactions, holds promise for the discovery of innovative treatment strategies. Interdisciplinary collaboration and knowledge sharing among researchers and clinicians, allowing efficient translational research, are crucial for addressing the global burden of tuberculosis.

## Figures and Tables

**Figure 1 biomolecules-13-00999-f001:**
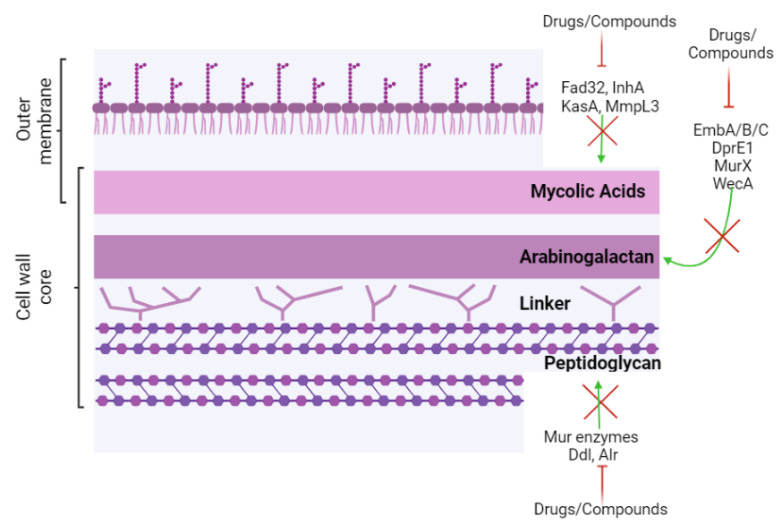
Schematic representation of the Mtb cell envelope with the principal targets for anti-TB compounds. (created by Biorender.com). The principal enzymes involved in the synthesis of different cell wall components are reported that are targeted by anti-Tb compounds.

**Figure 2 biomolecules-13-00999-f002:**
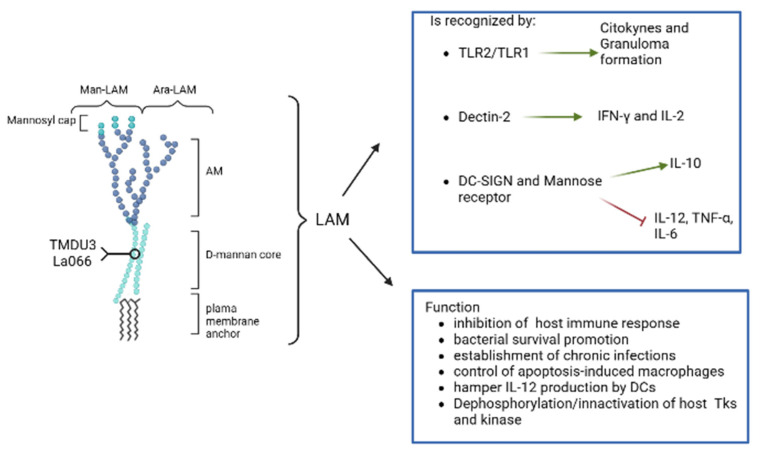
Lipolamman (LAM) structure, function and recognition by host immunity (created by Biorender.com).

**Figure 3 biomolecules-13-00999-f003:**
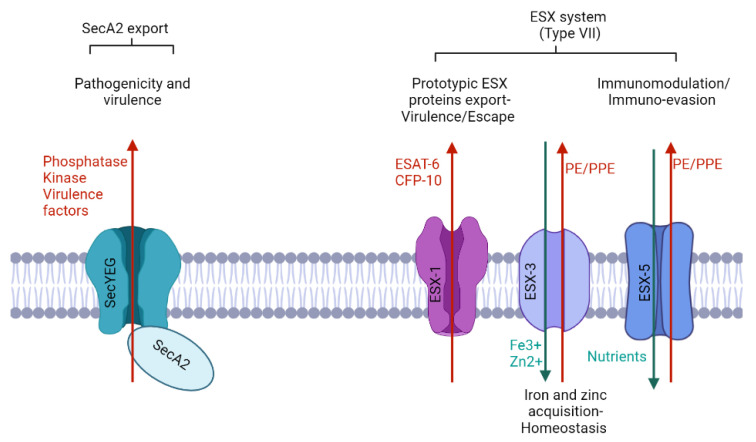
Schematic model of specialized mycobacterial export systems and their functions (created by Biorender.com). Each export system has a particular function which is reported schematically in the image: SecA2 is involved in pathogenicity and virulence; ESX-1 is the prototypic ESX protein export system and is involved in virulence and escape of Mtb; ESX-3 is involved in iron and zinc acquisition, and is responsible of homeostasis balance and in the mid-time acts as exporter for some PEs/PPEs, while ESX-5 is the major PE/PPE exporter and it is involved in nutrient uptake. In the image, red arrows indicate the export from Mtb bacillus, while the green ones represent the intake of bacteria.

**Table 1 biomolecules-13-00999-t001:** Chemical structures of molecules that are active on mycolic acids of the Mtb cell wall and their targets.

Target	Compound	Structure
Fad32	Diarylcoumarins [17] Pre-clinical	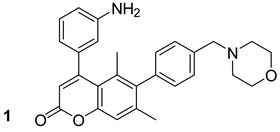
InhA	Isoniazid [15] First line drug	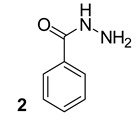
	1H-1,2,3 triazole-tethered quinoline-isoniazid conjugates [15] Pre-clinical	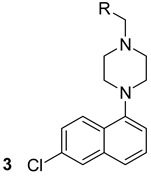
MmpL3	AU1235 [16] Phase I clinical trial	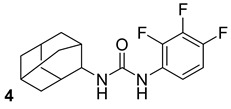
	BM212 [16] Phase I clinical trial	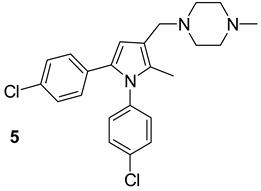
	SQ109 [16] Phase II clinical trial	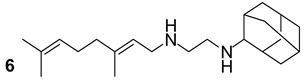
KasA	GSK3011724A [18] Phase II	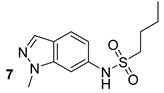
	(*E*)-1-(furan-3-yl)-3-(1H-indol-3-yl)prop-2-en-1-one [19] Pre-clinical	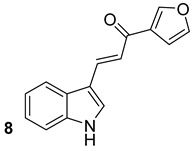
InhA/KasA /MmpL3	Ethionamide [20] Second-line drug	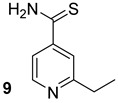
	Delamanid [21] Phase III clinical trial	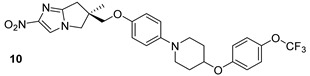
	Prothionamide [22] Second-line drug	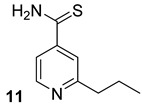
	Cytosporone E analogous [23] Pre-clinical	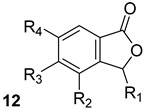

**Table 2 biomolecules-13-00999-t002:** Chemical structures of molecules active on arabinogalactan of the Mtb cell wall and their targets.

Target	Compound	Structure
DprE1	BTZ043 [16] Phase II clinical trial	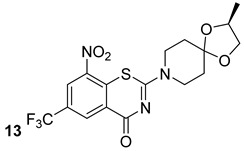
	BTZ169 [16] Phase II clinical trial	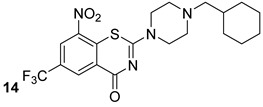
MurX	Caprazene [32] Pre-clinical	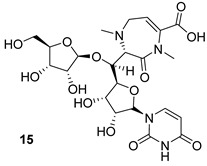
WecA	CPZENE-45 [32] Pre-clinical	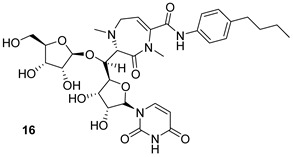

**Table 3 biomolecules-13-00999-t003:** Chemical structures of molecules active on peptidoglycans of the Mtb cell wall and their targets.

Target	Compound	Structure
Alr/Ddl	D-cycloserine [44] Second-line drug	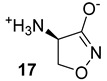
MurB	Sulfadoxine [46] FDA-approved drugs reproposed	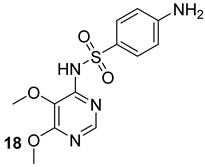
	Pyrimethamine [46] FDA-approved drugs reproposed	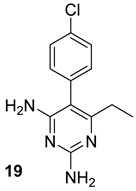
MurE	Sildenafil [46] FDA-approved drugs reproposed	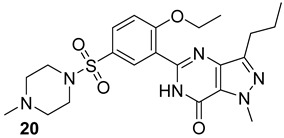
	Lifitegrast [46] FDA-approved drugs reproposed	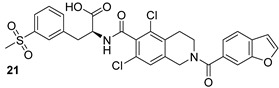
MurD	ZINC11881196 [47] Pre-clinical	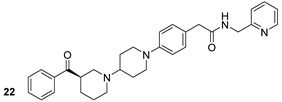
	ZINC12247644 [47] Pre-clinical	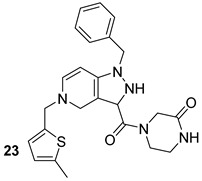
	ZINC14995379 [47] Pre-clinical	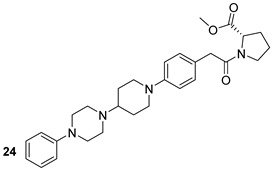
	PubChem6185 (digossina) [47] Pre-clinical	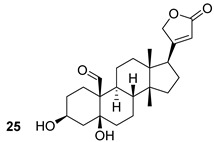

**Table 4 biomolecules-13-00999-t004:** Chemical structures of molecules active on different types of LAM in the Mtb cell wall and their targets.

Target	Compound	Structure
Mannan core of LAM	TMDU3 [55] Pre-clinical	anti-LAM monoclonal IgMs
	La066 [55] Pre-clinical	anti-LAM monoclonal IgMs
LAM biosynthesis	Ethambutol [17] First-line drug	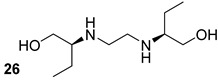

**Table 5 biomolecules-13-00999-t005:** Chemical structures of molecules active on different types of secretory systems in the Mtb cell wall and their targets.

Target	Compound	Structure
SecA2 ATPase component	Bengal analogs [65] Pre-clinical	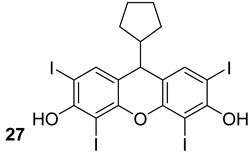
SecA2 Ion-channel and ATPase activities	SCA-15 [66] Pre-clinical	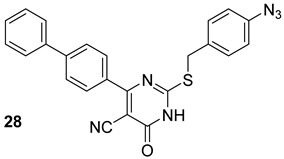
SecA2	Q30, M9, M26,U8,R26 Thiazolidone derivatives [67] Pre-clinical	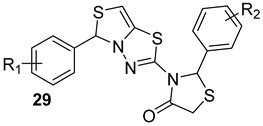
ESX-1 Block the secretion	Chloramphenicol [68] Pre-clinical	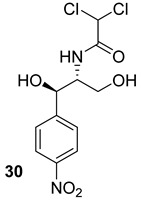
	Kanamycin [68] Second-line combinational drug	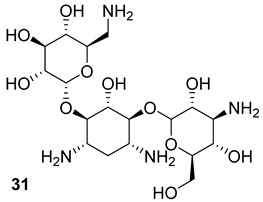
	Bortezomib [68] Pre-clinical	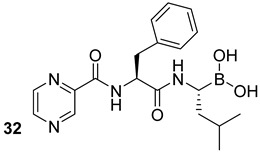
ESX-3 MbtI	5-(3-cyano-5-(trifluoromethyl) phenyl)furan-2-carboxylic acid [69] Pre-clinical	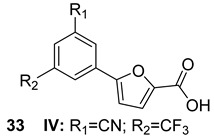
ESX-5 Blocks the export	1,2,4-oxadiazole scaffold [70] Pre-clinical	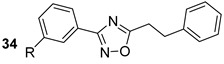

## Data Availability

No new data were created or analyzed in this study. Data sharing is not applicable to this article.

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
