# Peer review of "Emerging Extracellular Molecular Targets for Innovative Pharmacological Approaches to Resistant Mtb Infection"

_biomolecules, 2023, doi:10.3390/biom13060999_

Round 1

Reviewer 1 Report

The manuscript submitted by Francesco Peri et al reviewed existing and potential molecular targets from cell wall components, lipoarabinomannan, and the secretory systems of Mycobacterium tuberculosis (Mtb).

Overall, the manuscript is well written and the structure of the article is clear and logically presented. The figures are well prepared and facilitated the understanding of the content. Moreover, the chemical structures of pharmaceutical molecules against those molecular targets mentioned in the article facilitated the comparison between analogues blocking the same molecular target.

There are several mistakes or misleading wording in the article that will need to be revised:

Line 22: Contracting or contacting?

Line 125: Finally,

Line 139: replace “that” with “GalN”

Line 351: virulence factors also include secretory proteins (for example CFP10 and ESAT6)

Additionally, I have the following comments that could improve the manuscript:

·       In the abstract section, the authors wrote “highlighting the mechanism of action for each molecule”, however, some of the molecular inhibition mechanism is not clearly described (for example how thiazolidone derivatives inhibit the SecA2 pathway, how 5-(3-cyano-5-333 (trifluoromethyl) phenyl)furan-2-carboxylic acid inhibit the Salicylate synthase, how 1,2,4-oxadiazole-containing compounds inhibit ESX-5 secretion, etc, etc)

·       The title of this manuscript is “Emerging molecular targets for innovative pharmacological approaches to resistant Mtb infection”, however, there are also emerging molecular targets in the cytoplasm (for example DosS-DosT/DosR system) that were not mentioned in this article. Considering the molecular targets reviewed by author are all extracellular, it might be a good idea to specify the range of molecular targets discussed in this manuscript in the manuscript title.

·       The second paragraph and fourth paragraph of the conclusion section overlap. I recommend that the second paragraph concludes the manuscript while the fourth paragraph is used to discuss some future perspectives.

Author Response

We thank the reviewer for the observations and corrections that helped us in improving the readability and clarity of the manuscript:

  • We revised the manuscript and corrected reported errors and other typos
  • In the abstract we deleted the sentence: “highlighting the mechanism of action for each molecule”. As correctly outlined by this referee, some mechanisms of action (MOA) have been not reported in this review. The problem is that not all MOA are available from literature, so we were able to report only selected MOA relatively to some molecules.
  • Again, referee’s observation is totally correct. The title has been changed into: “Emerging extracellular molecular targets….”
  • The conclusion paragraph has been rewritten according to reviewer’s recommendation

Reviewer 2 Report

This review provides significant insights into the field of microbiology concerning M. tuberculosis infection. It highlights emerging pharmacological strategies that target the key virulence factors of antibiotic-resistant bacteria. Additionally, the review offers a comprehensive understanding of Mtb pathogenesis at the molecular level and suggests potential therapeutic strategies to address the escalating global threat of antibiotic-resistant Mtb infection.

The organization of the manuscript greatly facilitates reading by effectively addressing the crucial structures and functions underlying Mtb pathogenicity. However, it would be beneficial to clarify the type of review being presented. While it appears to be a narrative review due to the absence of inclusion and exclusion criteria, explicitly mentioning this information would enhance the manuscript.

Furthermore, Figure 1 lacks adequate contextualization, as its legend lacks necessary details. Additionally, the meaning of the green arrows in the figures is unclear, and the figures themselves appear to be cut off. Although this information may be implied, providing more explicit and detailed explanations would enhance readability.

The absence of information regarding the clinical phases in which the drugs/compounds are found in Table 1 is a notable oversight.

The conclusions effectively summarize the key findings and contribute to the reader's understanding of the current state of the field. This helps to elucidate the contemporary panorama surrounding the topic under investigation.

Author Response

We thank the reviewer for the observations and corrections that helped us in improving the readability and clarity of the manuscript:

  • As suggested, we stated at the beginning of the manuscript the aim and scope of the review:
  • “Aim of this review is to collect emerging pharmacological strategies for the development of drugs against Mtb infection deriving from the discovery of new targets. However, because the wideness of the topic, we limited our analysis to drugs/molecules targeting the cell wall polymeric components and the secretory proteins.”
  • Figure 1 has been improved (arrows should now be clearer) and more information has been added to the caption
  • Table 1 has been divided into 5 tables (one per class of compounds) to make the manuscript more readable and clinical phase of development has been added for each compound/drug